# Analysis of the Influence of Shrinkage Tensile Stress in Potting Material on the Anti-Overload Performance of the Circuit Board

**DOI:** 10.3390/s21072316

**Published:** 2021-03-26

**Authors:** Lei Sun, Wenjun Yi

**Affiliations:** National Key Laboratory of Transient Physics, Nanjing University of Science & Technology, Nanjing 210018, China; sunlei_jane@sina.com

**Keywords:** anti-overload, MEMS, potting adhesive, shrinkage stress, expansion stress

## Abstract

In this article, the influence of shrinkage tensile stress in potting materials on the anti-overload performance of a circuit board was studied. Firstly, the phenomenon of shrinkage tensile stress in common potting materials was analyzed, and it was found that the commonly used potting adhesives displayed large shrinkage characteristics. Secondly, a small experiment was set up to verify that the shrinkage tensile stress of potting adhesives would lead to printed circuit board (PCB) deformation, and the shrinkage stress was contrary to the acceleration direction of overload. Thirdly, the influence of potting adhesives on the overload resistance of the PCB was analyzed. However, the shrinkage tensile stress in the potting adhesive weakened the anti-overload ability of the circuit board. When there was a small amount of expansion stress in the potting adhesive, the overload resistance of the circuit board could be partially increased. From the analysis, it is indicated that a material with a certain expansion property, elasticity, and dense structure should be selected as the potting adhesive. This article provides a reference for improving the overload resistance of electronic devices.

## 1. Introduction

Micro-inertial navigation electronic devices have widespread application in the national economy and military fields. Especially in recent years, the combination of micro-inertial navigation electronic equipment and ammunition has not only miniaturized the guided ammunition but also saved on costs [1,2]. Navigation electronic devices will suffer various impacts in motion, especially when the guided projectile is launching. The load on the projectile will receive a strong instantaneous impact, and the maximum overload can reach tens of thousands of g [3]. Such a large overload may cause damage to the inertial and other electronic devices. The inertial device plays an important role in measuring the attitude and velocity of the projectile. Therefore, the overload resistance of the electronic components of the navigation system becomes the key constraint in the application of guided ammunition. Generally, to improve the impact resistance of missile-borne electronic systems, it is necessary to adopt certain protective measures, strengthening the protection of inertial measurement elements [4]. One effective way is to fill the remaining space inside the structure with buffer materials when the devices are assembled [5].

Many scholars have studied how to improve the anti-overload performance of devices, including rubber damping, potting resin adhesive, and other methods. Wang D. R. [6] designed a gyroscope shell with a soft elastic damping pad. Through numerical analysis, it was verified that the rubber damping pad suppressed the impact to the device. Yang P. J. et al. [7,8] used the elastic strain energy function method to evaluate the characteristic curve of rubber materials. Chen Y. [9] calculated the dynamic response of the packaging structure with or without a rubber cushion, which showed that the damping cushion protected the packaging structure better. The research of the above scholars shows that rubber damping has a good effect.

In addition, according to the research of some scholars [10,11], potting resin adhesive may also be a suitable method. The common potting materials are epoxy resin, polyurethane, etc. Lv C. Q. et al. [12] investigated the packaging status of an accelerometer under a high-overload environment and found that epoxy resin has a good effect as a potting adhesive. Zheng C. et al. [13] suggest that polyurethane can protect circuit structures from high impact through simulation. Li X. F. et al. [14] tested the vibration characteristics of polyurethane with different densities. Xia A. L. [15] introduced the process of encapsulating electronic equipment with rigid polyurethane foam material. Zhang K. et al. [16] used the liquid-crystal group to enhance the performance of self-made epoxy resin. The research of the above scholars shows that potting resin adhesive is a good method to improve overload resistance.

However, the parameters and curing process of the potting materials also have different effects on the high-overload resistance. Xu X. et al. [17] analyzed the influence of various parameters of potting materials on the response of electronic devices and put forward an optimization scheme for different engineering requirements. By analyzing the buffering mechanism of electronic circuit potting materials under high impact, An C. J. et al. [18] concluded that the potting materials must have a good energy-absorbing effect and fast attenuation of stress waves. Based on the energy-absorbing mechanism and curing method of the potting materials, Jiao M. [19] suggests that choosing an appropriate potting process is an effective way to improve the overload resistance of the circuit. Li C. et al. [20] found that the overload capacity of the sensor can be improved by choosing a potting adhesive with a relatively large elastic modulus and low density.

From the research of the above scholars, it can be seen that using a potting adhesive is an effective method to improve anti-high-overload ability. At present, polyurethane and epoxy resin are commonly used in electronic products under a high-overload environment. Polyurethane and epoxy resin materials have the advantages of high elasticity, strength, and insulation. Their formulas can also be changed for different densities, hardness, and strengths to achieve better performance. From the above analyses, the overload resistance of circuit devices is closely related to the level of assembly and packaging [21]. Some devices themselves have strong overload resistance, but when the assembly is unreasonable, the overall anti-overload ability may still decrease. For example, in the experiment of Jin J. C., the overload resistance of ceramic substrate exceeded 5000 g, but the capacity after metal packaging was less than 3000 g [22]. This demonstrated that the high-overload resistance of the circuit after assembly had a strong relationship with the packaging method. The research in this article found that the improper use of potting materials can reduce the overload resistance, especially the shrinkage stress formed after the filling material solidifies, which will weaken the overload resistance. However, there are few reports on the effect of shrinkage stress on the overload resistance of potting adhesives. This article will focus on the impact of shrinkage stress on the anti-overload performance.

## 2. Curing Process of the Potting Material

Encapsulation is a type of non-porous, complete packaging technology for electronic device systems. Liquid potting materials such as epoxy resin are poured into the device by mechanical or manual methods to cure into the polymer insulation material, with excellent thermosetting performance under normal temperatures or heating conditions. When the impact pressure is generated, the elastic–plastic deformation and damping effect of the materials can absorb energy. The stress–strain curve of the potting material includes three stages: elastic deformation, yield plateau, and material compaction. The material has to go through a long yield plateau before compaction, which determines the buffering and energy-absorbing properties of the potting material.

It has been found that stress concentration is a common problem that occurs in the potting body, and it may produce an effect on the circuit system. Generally speaking, epoxy resin and other potting materials mainly produce curing shrinkage stress, thermal stress, and external applied stress during curing [23]. Among these, shrinkage stress includes chemical shrinkage in the process from liquid to solid and physical shrinkage in the cooling process [24], which is an inevitable state in the processing of composite resin materials. When the composite resin is cured, it transforms from monomers to network macromolecules, from colloidal to solid, resulting in volume shrinking. In general, the volume of the oligomer resin will shrink more or less during the curing process. The volume shrinkage range of common resin materials is between 1.5% and 3%. However, some epoxy resins’ shrinkage range is even higher than 5%. This high shrinkage rate will lead to shrinkage stress inside the resin, which easily causes internal stress to converge to a point, becoming a potentially destructive factor. Shrinkage stress may also result in a sharp reduction in the material strength and even cause material cracking [25]. In the case of reciprocating bending, the PCB and the potting material will have different deflections; the solder joint of the board connection may experience repeated tension and compression, and this will eventually lead to chip fracture [26]. Therefore, it is necessary to analyze the effect of shrinkage stress of potting materials on the overload resistance of circuit boards.

According to the mechanics of materials, shrinkage stress can be expressed as:σ_s_ = *ε*_s_ × *E*_s_,(1)
where σ_s_ is the curing shrinkage stress, *ε*_s_ is the shrinkage strain of cured resin, and *E*_s_ is the elastic modulus of resin. As in Formula (1), the shrinkage stress depends on the volume shrinkage of the resin and the elastic modulus. The greater the shrinkage deformation of resin is, the larger the shrinkage stress is, which may lead to the deformation of the PCB and affect its anti-overload capacity. Reducing the elastic modulus of the resin can reduce the shrinkage stress. However, the reduction of the elastic modulus will lead to the failure of the PCB obtaining effective support, which will also reduce the overload resistance. Here, we carry out an experiment to analyze the impact of shrinkage stress on the PCB.

## 3. Experimental Analysis of Shrinkage Stress of Potting Material

### 3.1. Introduction of the Experiment

In order to determine the influence of shrinkage stress on the overload resistance of the PCB, we designed a simple experiment to simulate the packaging state of the PCB. It used an aluminum square shell as a support; the inside of the shell was filled with common potting materials, and the PCB was placed on the hollow section of the shell. A full bridge resistance strain gauge was pasted onto the PCB to measure the stress influenced by the potting material’s curing. A schematic diagram of the experimental device is shown in Figure 1. The potting adhesive was located at the rear of the PCB along the acceleration direction (such as the lower part in Figure 1), to buffer the inertial force caused by the impact of acceleration. Moreover, the potting adhesive could not be placed in the upper part of the PCB as the potting adhesive would have produced a huge inertial force and directly damaged the PCB. The potting adhesive used here was a common two-component polyurethane. Polyurethane material has low hardness, good elasticity, strong adhesion, and good insulation, and it is commonly used in the encapsulation of inertial guidance in electronic devices.

Firstly, we pasted the strain gauge onto the inside of the PCB, as shown in Figure 2a, placed the wires at the back, as shown in Figure 2b, and fixed the aluminum square shell ont**o** the thick base with glue. Secondly, the potting adhesive was prepared and poured into the aluminum square shell, and the shell was vibrated to discharge the bubbles, as shown in Figure 2c. Thirdly, we placed the PCB above the aluminum square shell and connected the signal acquisition wire. The data acquisition interface is shown in Figure 2d. Finally, we recorded the data when the potting adhesive was solidifying at room temperature. The cross-sectional dimension of the shell was 40 mm × 40 mm, and the PCB was 48 mm × 48 mm.

### 3.2. Strain Analysis of the Circuit Board during the Curing of the Potting Material

The curing test of the potting material was carried out twice at room temperature, and these were recorded as Test 1 and Test 2. The aluminum shell and the PCB were the same size in the two tests. The strain trend measured during the curing process is shown in Figure 3. It can be seen that the curves of strain show a law of quadratic linear change. The strain in the negative direction of the PCB changed rapidly within 600 min, and then the rate of strain reduction gradually slowed down. At this time, the potting adhesive gradually solidified. A negative strain indicates that the PCB was stretched inward due to the shrinkage of the potting adhesive, and the maximum was −3297.6 με. Both tests verified that the PCB was subjected to large tensile stress due to the shrinkage of the potting adhesive, and the direction of this shrinkage tensile stress was consistent with the inertial force, which made the PCB more easily damaged.

In order to further explore the influence of shrinkage stress on PCB deformation, a comparative experiment was carried out, in which three types of weights were placed on the PCB to analyze the strain of the PCB under the pressure of the weights. The masses of the three weights were 200 g, 284 g, and 633 g. The weight was placed at the center, above the PCB, and the PCB center deformed downward under the action of gravity. If the strain is negative and continues to increase, this indicates that the force exerted by the weight and the shrinkage stress are in the same direction, and the PCB deformation is also in the same direction. The curves of strain variation are shown in Figure 4.

The changes in strain in Figure 4 show that the deformation direction of the PCB caused by weight gravity is consistent with that caused by the potting adhesive’s shrinkage stress, and the strain of the PCB is around −728.1 με when the weight gravity is 633 g. This phenomenon verifies that the shrinkage stress occurs contrary to the acceleration direction, which indicates that the PCB may receive a certain deformation in the direction of acceleration before launch. At the same time, it can be seen from the results that the shrinkage stress and strain caused by the potting adhesive are relatively large, able to reach the effect of loading hundreds of grams.

### 3.3. Simulation Analysis of Overload on the Circuit Board

We established a three-dimensional numerical model of the device. The shell frame was set as aluminum, the size of which was around 40 mm × 40 mm × 40 mm; the upper part and the side of the shell were made of three PCBs, and the inner part was filled with polyurethane. The equivalent stress and deformation of the inertial electronic devices under acceleration overload impact were calculated by the finite element method in the ANSYS environment; the results are shown in Figure 5.

The direction of the overload in the vertical line had a value from 0.1 g to 3.5 × 4 g, and the bottom plane of the shell frame was fixed. We calculated stress in the PCB under the conditions wherein the potting material was either filled or not. The results are shown in Figure 6. The green curve represents the stress in the PCB with the change in the acceleration value when the potting adhesive was filled, and the red curve represents the stress changes when the potting adhesive was not filled.

The calculation results show that the filling of the potting adhesive has significant effects on reducing the equivalent stress of the circuit board. If there was no potting adhesive filled between the circuit board and the frame, the circuit board was in a suspended state, and it was easier to deform and produce increased stress. From the curves, when the potting adhesive was not filled, the stress in the circuit board reached 32.1 MPa at 20,000 g, but the stress in the circuit board was only around 16.8 MPa when the potting adhesive was injected, which reduced the stress by 47.7%.

### 3.4. Influence of Shrinkage Stress on the PCB’s Overload Resistance

The previous analysis shows that filling the potting adhesive is an effective way to improve the overload resistance of the PCB. However, further analyses found that the shrinkage stress of the potting adhesive may lead to a significant decrease in the overload resistance of the PCB. Most of the invalidity in circuit boards is due to the different elastic modulus values of the PCB and the components welded onto it, which causes a different type of deformation in each component and leads to solder joint fracture. Therefore, to improve the overload resistance of navigation electronic devices, it is important to control the deformation of the PCB.

Figure 7 shows the equivalent stress and the total deformation of the PCB, with shrinkage stress inside, when the overload is 10,000 g, 20,000 g, and 30,000 g. The plane of this PCB is perpendicular to the overload direction. It is obvious that the equivalent stress and the total deformation of the PCB increase with the increase in shrinkage stress at different overloads. Specifically, the greater the shrinkage stress produced by the potting adhesive, the weaker the overload resistance of the PCB. In the calculation, we simulated the material shrinkage stress by applying a tensile stress to the inner surface of the PCB and frame.

If the composition of the potting adhesive is controlled so that it does not produce shrinkage stress but produces expansion push stress, how will the PCB be affected?

Figure 8 shows the equivalent stress and the total deformation of the PCB, with expansion stress inside, when the overload is 10,000 g, 20,000 g, and 30,000 g. The equivalent stress in the PCB decreases with the increase in the expansion stress generated by the potting adhesive. However, under 10,000 g overload, when the expansion stress is greater than 1.5 MPa, the equivalent stress increases with the rise in expansion stress. This is because when the expansion stress is too large, exceeding the demand of overload resistance, the expansion stress will increase the stress in the PCB. In other words, the expansion stress can reduce the stress in the PCB, but the expansion stress should not be too large. In Figure 8b, the total deformation of the PCB decreases with the increase in expansion stress.

Compared with Figure 7a and Figure 8a, the overload resistance of the PCB, which is perpendicular to acceleration, is affected by the internal stress of the potting adhesive. Taking 20,000 g overload as an example, when the shrinkage stress is 2 MPa, the equivalent stress in the PCB is 41.2 MPa, which is almost 44.6% higher than that without internal stress. Regarding the expansion stress, the equivalent stress in the PCB is 14.6 MPa, which is almost 46.3% lower than that without internal stress.

For further analysis, the influence of the internal stress on the PCB which is parallel to the overload direction was also studied.

Figure 9 shows the equivalent stress of the PCB, with shrinkage stress inside, under three different overloads. The plane of this PCB is parallel to the overload direction. In Figure 9a, the equivalent stress rises slowly with the increase in shrinkage stress under 20,000 g and 30,000 g. When under 10,000 g, when the shrinkage stress is less than 1.75 MPa, the equivalent stress also increases slowly, but when it is more than 1.75 MPa, the increase becomes larger. In Figure 9b, when under 20,000 g and 30,000 g overload, and 10,000 g where the expansion stress is less than 1.75 MPa, the equivalent stress in the PCB decreases slowly with the increase in expansion stress. Moreover, when the expansion pressure is greater than 1.75 MPa under 10,000 g overload, the equivalent stress increases slowly with the increase in expansion stress.

The results demonstrate that the anti-overload ability of the PCB which is parallel to the overload direction is affected by the internal stress of the potting adhesive. With the increase in shrinkage stress, the anti-overload ability decreases, and with the increase in expansion stress, the anti-overload ability increases. However, when the expansion stress is too large, it will still cause significant stress in the PCB. Moreover, the effect of this internal stress on the PCB in the parallel direction is much smaller than that in the perpendicular direction.

Figure 10 shows the total deformation of the PCB which is parallel to the overload direction under three overloads. It can be seen that regardless of whether the internal stress of the potting adhesive is shrinkage stress or expansion stress, the total deformation of the PCB increases slowly with the increase in internal stress. However, it is noted that the total deformation of the PCB parallel to the overload direction is one order of magnitude less than that of the PCB which is perpendicular to the overload direction, in Figure 7b and Figure 8b.

Figure 11 and Figure 12, respectively, show the influence of internal stress of the potting adhesive on the equivalent stress and the total deformation of the metal frame under three overloads. It can be seen from Figure 11a and Figure 12a that when the internal stress is shrinkage stress, the equivalent stress and the total deformation of the metal frame both increase with the increase in shrinkage stress, although the increase is not greater than that of the PCB which is perpendicular to the overload direction. It can be clearly seen from Figure 11b and Figure 12b that when the internal stress is expansion stress, the equivalent stress and the total deformation of the metal frame decrease slowly with the increase in expansion stress. In summary, when the internal stress of the potting adhesive is shrinkage stress, the overload resistance of the metal frame decreases with the increase in shrinkage stress, and the overload resistance of the metal frame increases with the increase in expansion stress.

The above analyses show that the shrinkage stress of the potting adhesive will reduce the overload resistance of electronic devices, especially when the PCB is perpendicular to the overload direction, and appropriate expansion stress is beneficial to improve the overload resistance. However, the expansion stress of the potting adhesive should not be too large. Excessive expansion stress will lead to PCB deformation and will also damage the circuit board. The research in this article demonstrates that a material with suitable expansion, elasticity, and density should be selected for the potting adhesive.

## 4. Conclusions

This article carried out research on the influence of potting adhesives’ shrinkage internal stress on the overload resistance of a circuit board. Firstly, the shrinkage stress phenomenon of commonly used potting adhesives was analyzed. It was found that potting adhesives always have a shrinkage rate. Secondly, an experiment was set up to determine whether the shrinkage tensile stress of the potting adhesive would cause PCB deformation. It showed that the direction of shrinkage tensile stress is contrary to the acceleration direction of overload, which may reach −3297.6 με. Thirdly, a simulation model was established to analyze the influence of potting adhesives on the overload resistance of the PCB. The anti-overload ability of the PCB was greatly improved when it was filled with potting adhesive, which could be increased by 47.7% at 20,000 g. However, the shrinkage tensile stress produced in the potting adhesive weakened the anti-overload ability of the circuit board. When the shrinkage stress was 2 MPa, the equivalent stress on the PCB was 41.2 MPa, which was almost 44.6% higher than that without internal stress under the condition of 20,000 g. It was also found that when there was a small amount of expansion stress inside the potting adhesive, the overload resistance of the circuit board could be partially increased. The equivalent stress of the PCB was 46.3% lower than that without internal stress under 20,000 g overload, at an expansion stress of 2 MPa. However, the inflection points in Figure 8a and Figure 9b indicate that the expansion stress should not be too large and must be controlled within a certain range.

Through the analyses of this article, it is indicated that a material with certain expansion, elasticity, and density should be selected when filling the potting adhesive. This article may provide a reference for improving the anti-high-overload ability of electronic devices.

## Figures and Tables

**Figure 1 sensors-21-02316-f001:**
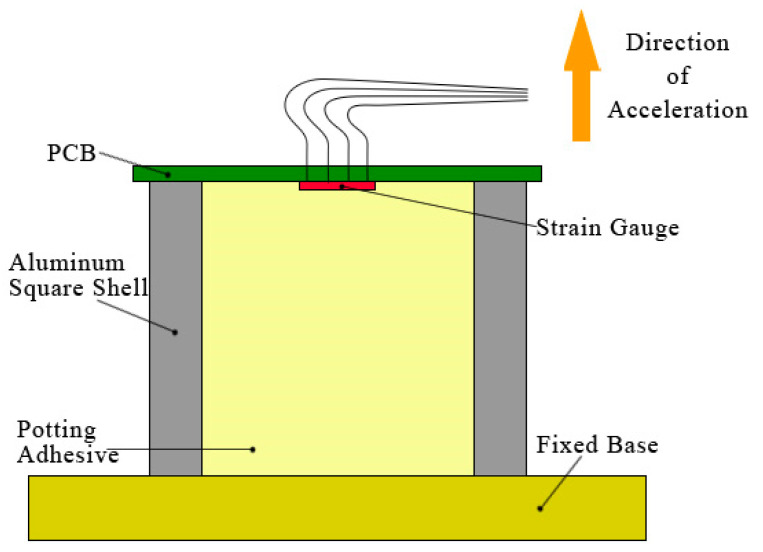
Schematic diagram of the testing device.

**Figure 2 sensors-21-02316-f002:**
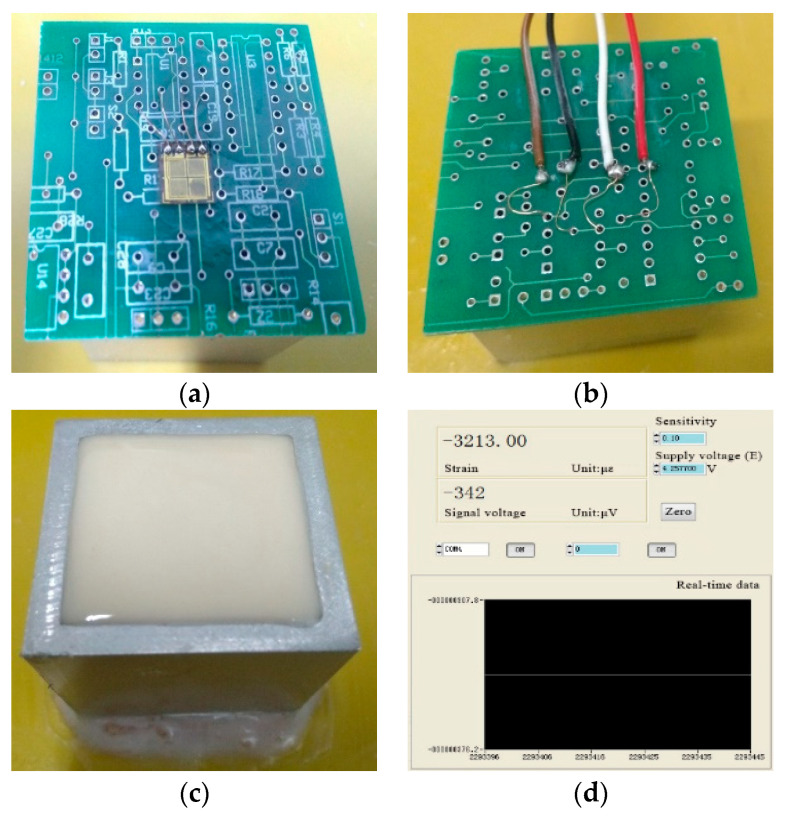
Test device for shrinkage stress of the potting adhesive: (**a**) Strain gauge on the printed circuit board (PCB); (**b**) Wires on the PCB; (**c**) Potting adhesive inside the test device; (**d**) Data acquisition interface.

**Figure 3 sensors-21-02316-f003:**
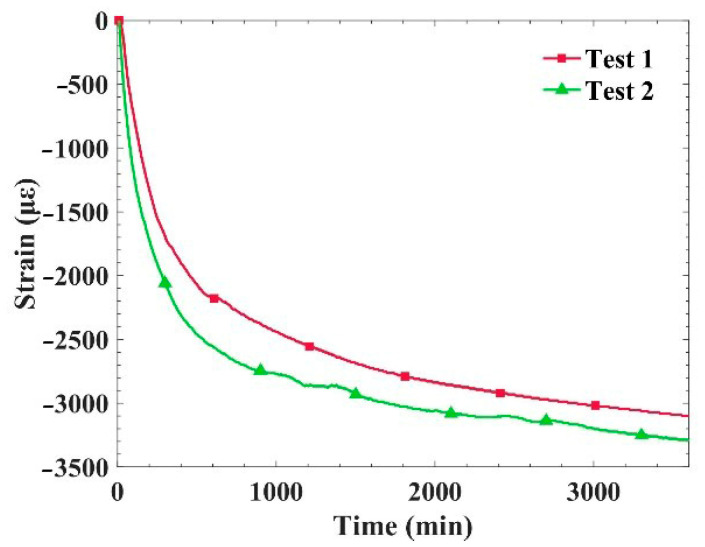
The strain of the PCB during the curing process of the potting adhesive.

**Figure 4 sensors-21-02316-f004:**
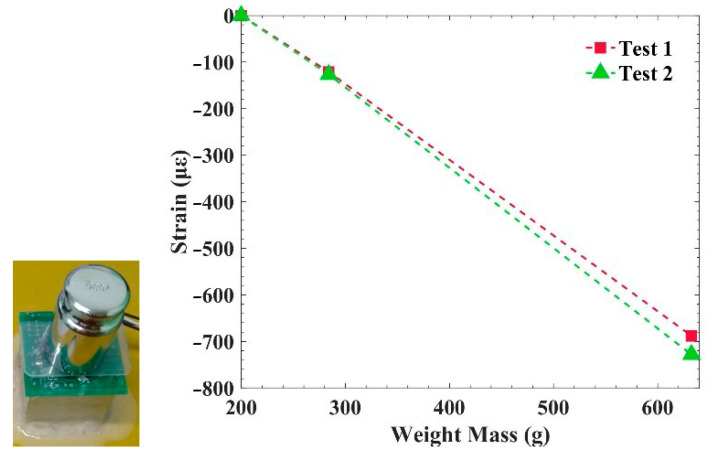
Influence of additional force on PCB strain.

**Figure 5 sensors-21-02316-f005:**
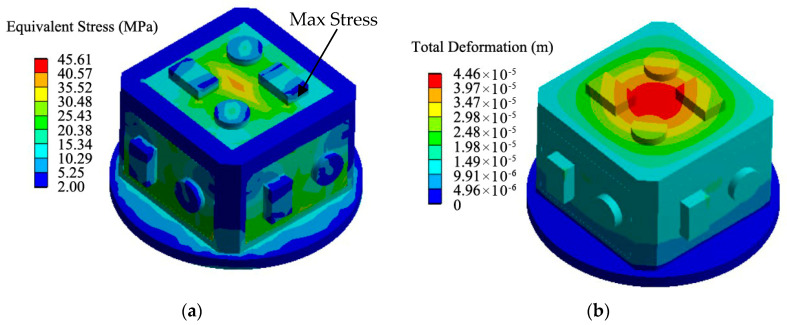
Three-dimensional stress analysis diagram of the test device: (**a**) Equivalent stress; (**b**) Total deformation.

**Figure 6 sensors-21-02316-f006:**
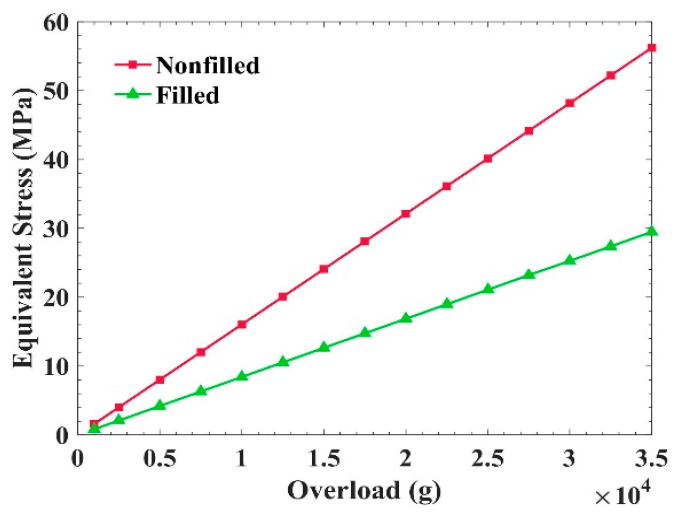
Stress changes in the PCB in the direction of vertical overload.

**Figure 7 sensors-21-02316-f007:**
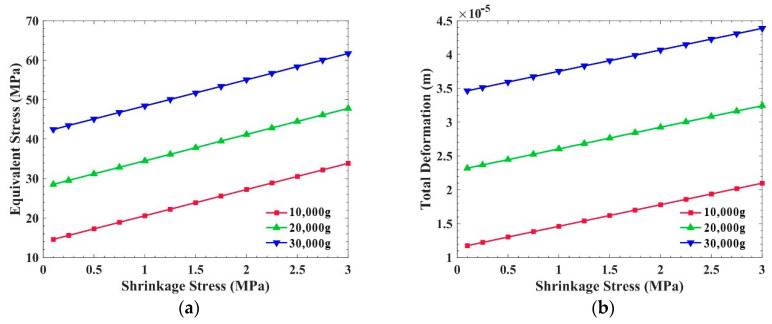
Influence of shrinkage stress on the PCB, which is perpendicular to the overload direction: (**a**) Equivalent stress of the PCB; (**b**) Total deformation of the PCB.

**Figure 8 sensors-21-02316-f008:**
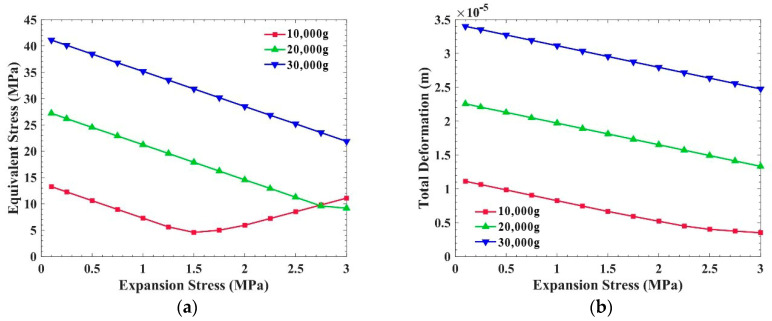
Influence of expansion stress on the PCB, which is perpendicular to the overload direction: (**a**) Equivalent stress of the PCB; (**b**) Total deformation of the PCB.

**Figure 9 sensors-21-02316-f009:**
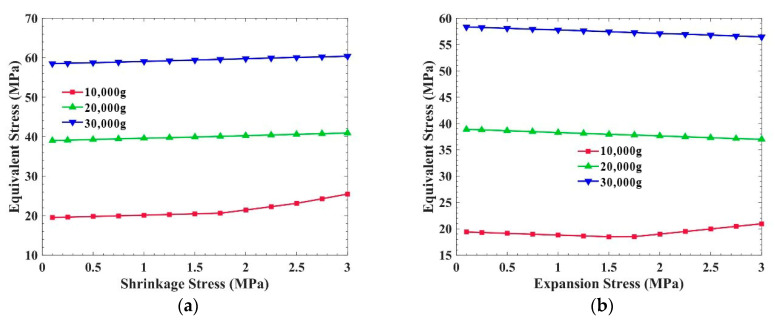
Influence of the internal stress of the potting adhesive on the equivalent stress of the PCB, which is parallel to the overload direction: (**a**) Shrinkage stress; (**b**) Expansion stress.

**Figure 10 sensors-21-02316-f010:**
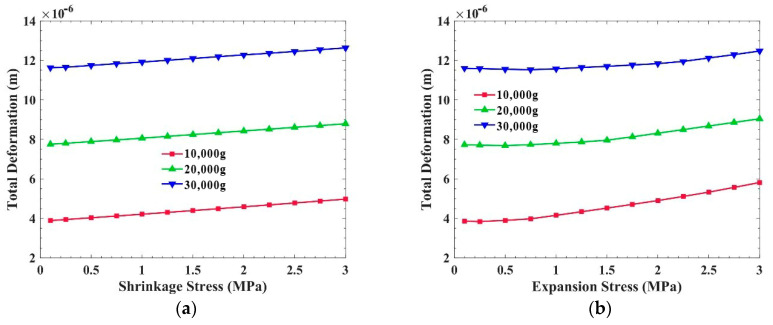
Influence of the different internal stresses of the potting adhesive on the total deformation of the PCB which is parallel to the overload direction: (**a**) Shrinkage stress; (**b**) Expansion stress.

**Figure 11 sensors-21-02316-f011:**
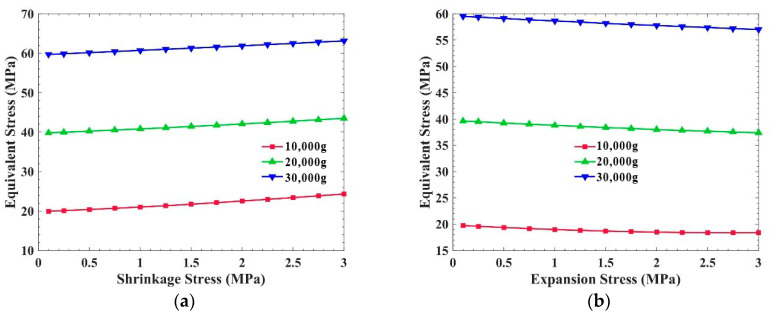
Influence of the different internal stresses of the potting adhesive on the equivalent stress of the metal frame: (**a**) Shrinkage stress; (**b**) Expansion stress.

**Figure 12 sensors-21-02316-f012:**
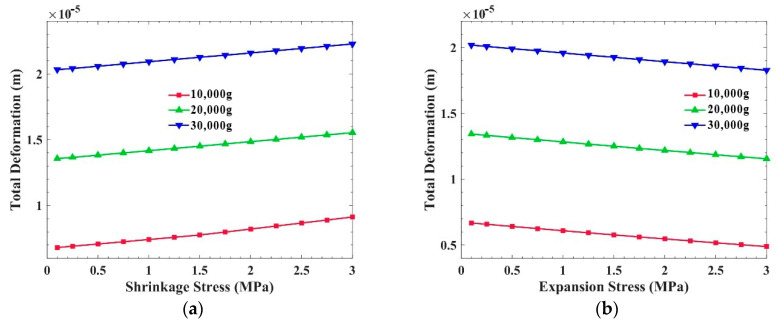
Influence of the different internal stresses of the potting adhesive on the total deformation of the metal frame: (**a**) Shrinkage stress; (**b**) Expansion stress.

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
