# Peer review of "Analysis of the Influence of Shrinkage Tensile Stress in Potting Material on the Anti-Overload Performance of the Circuit Board"

_sensors, 2021, doi:10.3390/s21072316_

Round 1

Reviewer 1 Report

The paper compares the effect of a potting material (bi-component polyurethane) residual stress on the mechanical behavior of a printed circuit board under an acceleration load of several tens of thousands of g. The comparison is carried out through experimental measurements on an ad hoc test specimen and numerical finite element simulations.

While the topic and the outcomes are interesting, the paper lacks some qualities for the publication. I encourage the authors to improve it according to some suggestions below.

First, the English is poor at several points: there are some nouns/verbs that are repeated one after the other (e.g. line 22 “guidance significance”, line 28 “applicate prospect”, then I stop here picking examples), possibly because of lack of final editing (see also the title of subsection 3.1, or line 51 “rubber cushion or else”); when a relative sentence is introduced, the verb is sometimes missing (e.g. “which IS”, at lines 245, 258, 279, 285, 303, 308); sometimes the speaking subject changes to the second person (e.g. lines 162-169); or the text is completely unedited, such as lines 210-214. Obviously, in this way the reading is difficult.

The Introduction part at lines 43-83 appears too dispersive and it does not help the reader to enter into the problem.

Why a “small” experiment? It is an experiment or not? However, I don’t believe you have enough points to connect them with lines in Figure 4.

Equation 1 is the Hooke’s law. When describing it, the term “deformation” should be replaced by “strain”, with a different symbol (εs). It is not just a matter of taste, since you use elsewhere the term “deformation” as a synonym of displacement, see Figures 5b, 7b, 8b, 10, 12, and in all of these figures the unit is (correctly) meters. The strain in the Hooke’s law is instead non dimensional, as in your Figure 3 or elsewhere in the text.

Line 135: “…from the above formula the shrinkage stress depends on the shrinkage rate of rate curing and the elastic modulus…” (emphasis is mine). Why the “rate”? Do you mean that all the quantities vary with time? Then you should modify accordingly equation 1.

You do not provide quantitative indications for Es (elastic modulus). Why?

What is the constitutive mechanical model adopted for the polyurethane in the simulations? How did you model the shrinkage stress? Is it an initial stress calculated apart and then assigned as an initial value to the potting material domain? The problem seems statically not determined, so the way you impose, for example, an initial strain is relevant.

Are all the other materials in the model (e.g. for PCB, aluminum frame or the base) elastic?

I don’t understand clearly how the parts (potting material, PCB, fixed base) are connected in the numerical model: do you consider perfect continuity between them? Are there possible contacts? I believe that the base is fixed, but you do not state clearly the boundary conditions.

Exactly, how did you apply the acceleration to the model? Is it an equivalent static load or an applied motion? Is there a time period for a half sine pulse with the given amplitude in g? Explain, please.

About the quantitative evaluation of stresses. I don’t see clearly from Figure 5a where the maximum stress is. I wonder if it is in a location where a stress concentration arises (due to geometry details such as rounded corners): in that case, depending on whether the model is elastic or not, then the quantitative estimation could be mesh dependent. If so, your comparison still holds true, but not in a quantitative sense; in particular, the minimum values in the Figures 8a, 9 could shift.

Reviewer 2 Report

The article submitted for review shows the impact of the shrinkage stress of the potting material on the anti-overload performance of electronic devices. The authors presented the current state of research on the overload resistance of electronic devices in a complete way. In an experimental way, the forces acting on the circuit board during curing of potting material were examined and through simulations, the influence of a potting material on circuit board anti-overload performance was tested.The article is very interesting and has great scientific value, but there are a few things that the authors need to correct before publication:

Major errors:
- English at article should be revised in terms of vocabulary and punctuation - In paragraph 1 of the introduction, should be a lot more citations confirming statements in the text
-The simulation environment and the parameters of simulation should be described in the article
-A deeper commentary on the obtained simulation results is required (preferably supported by citations), which shows the nature of the curve, why there is an inflection point, etc.

Minor errors:
-Picture 1 should be 20-30% larger
-Picture 2 should be on one page
-Formula (1) should have a citation
-Chapter 3 title should be on a new page

Round 2

Reviewer 1 Report

The paper improved in every part with this version. The authors' answers are enough for me.

Author Response

Dear Editors and Reviewers, Thank you for your kind comments on our manuscript entitled “Analysis of the influence of shrinkage tensile stress in potting material on the anti-overload performance of the circuit board” (Manuscript Number: sensors-1115520). I’m very happy that the modification and answers can satisfy you. Thank you again for your positive comments and valuable suggestions to improve the quality of our manuscript. Yours Sincerely

Reviewer 2 Report

The authors of the article have made almost all the corrections I have proposed, so I think the article is almost ready for publication. The only thing I think is necessary to add is an environment for modeling with the finite element method (is it SolidWorks, Autodesk Inventor, Ansys, Comsol or another?).
Track changes text helps to see what's changed in text, but in general, such text is hard to read. Authors, apart from text with corrections, should also include the text without underlined changes.

Author Response

Dear Editors and Reviewers, Thank you for your kind comments on our manuscript entitled “Analysis of the influence of shrinkage tensile stress in potting material on the anti-overload performance of the circuit board” (Manuscript Number: sensors-1115520). I’m very happy that the modification and answers can satisfy you. Point 1: It is necessary to add is an environment for modeling with the finite element method (is it SolidWorks, Autodesk Inventor, Ansys, Comsol or another?). Response 1: Thank you for your suggestion. I have rewritten the sentence “The equivalent stress and deformation of inertial electronic device under acceleration overload impact are calculated, which are shown in Figure 5.” to “The equivalent stress and deformation of inertial electronic device under acceleration overload impact are calculated by finite element method in ANSYS environment, which are shown in Figure 5.” in line 177-179. The added words “in ANSYS environment” are marked in red in the text and I hope it won't affect your reading. Thank you again for your positive comments and valuable suggestions to improve the quality of our manuscript. Yours Sincerely